# The orphan ligand, activin C, signals through activin receptor-like kinase 7

Erich J Goebel[1], Luisina Ongaro[2], Emily C Kappes[1], Kylie Vestal[1], Elitza Belcheva[3], Roselyne Castonguay[3], Ravindra Kumar[3], Daniel J Bernard[2], Thomas B Thompson[1]*

[1]Department of Molecular Genetics, Biochemistry, and Microbiology, University of Cincinnati, Cincinnati, United States; [2]Department of Pharmacology and Therapeutics, Centre for Research in Reproduction and Development, McGill University, Montreal, Canada; [3]Merck &Co., Inc., Kenilworth, United States

**Abstract** Activin ligands are formed from two disulfide-linked inhibin β (Inhβ) subunit chains. They exist as homodimeric proteins, as in the case of activin A (ActA; InhβA/InhβA) or activin C (ActC; InhβC/InhβC), or as heterodimers, as with activin AC (ActAC; InhβA:InhβC). While the biological functions of ActA and activin B (ActB) have been well characterized, little is known about the biological functions of ActC or ActAC. One thought is that the InhβC chain functions to interfere with ActA production by forming less active ActAC heterodimers. Here, we assessed and characterized the signaling capacity of ligands containing the InhβC chain. ActC and ActAC activated SMAD2/3-dependent signaling via the type I receptor, activin receptor-like kinase 7 (ALK7). Relative to ActA and ActB, ActC exhibited lower affinity for the cognate activin type II receptors and was resistant to neutralization by the extracellular antagonist, follistatin. In mature murine adipocytes, which exhibit high ALK7 expression, ActC elicited a SMAD2/3 response similar to ActB, which can also signal via ALK7. Collectively, these results establish that ActC and ActAC are active ligands that exhibit a distinct signaling receptor and antagonist profile compared to other activins.

## Editor's evaluation

The function of Activin C is poorly understood. The authors show that Activin C stimulates SMAD2/3 signaling via the type I receptor ALK7. However, Activin C binds the cognate Activin type II receptor with a lower affinity than Activin A or Activin B and is resistant to the extracellular antagonist Follistatin. Collectively, these data clarify the biological activity of Activin C and provide an important foundation for further research on Activin signaling.

*For correspondence: tom.thompson@uc.edu

## Introduction

The activins are multifunctional secreted proteins that play critical roles in growth, differentiation, and homeostasis in a wide variety of cell types. As part of the greater TGFβ family, the activins are dimeric in nature and built from two inhibinβ (Inhβ) chains of approximately 120 amino acids (e.g. activin A [ActA] is built from two InhβA chains) that are tethered by a disulfide bond. Members of the activin class include ActA, activin B (ActB), activin C (ActC), and activin E (ActE), and extend to include GDF8 (myostatin) and GDF11. The Inhβ chains share high sequence identity, such that InhβA and InhβB are 63% identical, with InhβC ~50% identical to both InhβA and InhβB (*Thompson et al., 2004*). In addition to homodimer formation, several combinations of heterodimers have been observed, such as activin AB (ActAB) formed between InhβA and InhβB chains, as well as the heterodimer activin AC (ActAC) comprises InhβA and InhβC chains (*Nakamura et al., 1992*; *Mellor et al., 2000*; *Mellor*

*et al., 2003*; *Butler et al., 2005*). While heterodimers can form, most studies have focused on the homodimeric forms of the ligands. In addition, due to their established biological roles, many studies have focused on characterizing the ligands ActA and ActB; however, few studies have characterized the ligands ActC or ActE, especially regarding their ability to signal.

The InhβC subunit was first identified from a human liver cDNA library (*H tten et al., 1995*). Its biological role was initially unknown due to the absence of hepatic phenotypes in InhβC knockout mice (*Lau et al., 2000*). Expression of InhβC is highest in the liver but has also been detected in reproductive tissues (*Gold et al., 2009*). InhβC has been proposed to function as an ActA antagonist, as coexpression of InhβA and InhβC results in the formation of the heterodimer ActAC, which is a less active signaling molecule than ActA (*Mellor et al., 2003*; *Gold et al., 2009*). For example, ActAC is less potent than ActA in IH-1 myeloma cells (*Olsen et al., 2020*). Thus, InhβC expression in the presence of InhβA not only reduces ActA levels but also forms the less potent ligand ActAC. It has also been proposed that ActC directly antagonizes ActA signaling (*Gold et al., 2009*). The mechanism for this is thought to be the binding of a non-signaling ActC to the ActA receptors, acting as a competitive inhibitor. These two mechanisms are similar to how inhibin-α (Inhα) forms heterodimers with InhβA chains to form inhibin A, reducing ActA production, and by competitively blocking ActA receptor binding (*Namwanje and Brown, 2016*). The similarities were confirmed through studies which showed that in Inhα knockout mice, which develop female reproductive tumors and have abnormally high levels of ActA resulting in cachexia, the ActA levels can be suppressed by over-expression of InhβC (*Gold et al., 2013*; *Bi et al., 2016*). While almost all studies have suggested an inhibitory role of InhβC, a recent study showed that ActC relieved chronic neuropathic pain in mice and rats,

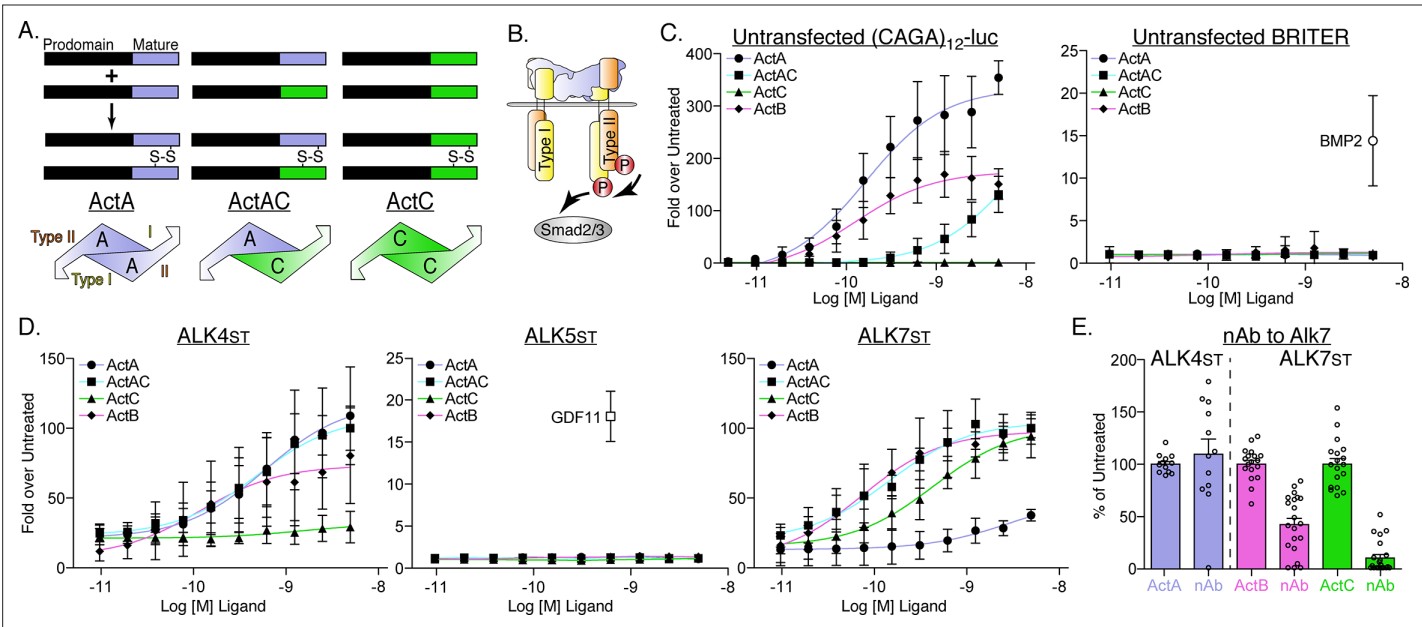

**Figure 1.** Differences in type I receptor utilization by ActA, ActAC, and ActC. (**A**) Schematic displaying formation of activin A (ActA), activin AC (ActAC), and activin C (ActC) from dimerization of inhibinβA (blue) and inhibinβC subunits (green). (**B**) Generalized TGFβ signaling schematic displaying activin-SMAD2/3 signaling with type II (orange) and type I (yellow) receptor binding positions displayed for ActA. (**C**) Luciferase reporter assay in response to ActA, ActAC, or ActC titration in untransfected (CAGA)$_{12}$-luciferase (luc) or BRITER HEKT cells. BMP2 was included as a positive control for the BRITER reporter. (**D**) ActA, ActAC, ActC, and ActB activation of (CAGA)$_{12}$-luc HEK293T cells transfected with SB-431542-resistant (Ser to Thr, ST) type I receptors. In (C) and (D), each data point represents the mean ± SD of triplicate experiments measuring relative luminescence units (RLU). ALK4$_{st}$ and ALK7$_{st}$ transfection assays in (D) were normalized to 100-fold from mean of highest point. EC50 values are reported in *Supplementary file 1*. (**E**) Effects of an ALK7 neutralizing antibody (nAb) on ActA, ActB, and ActC induction of (CAGA)$_{12}$-luc activity in cells expressing the indicated type I receptors. In (E), each data point represents a technical replicate within triplicate experiments with bars displaying the mean ± SD. In both (**D**) and (**E**), cells were treated with 10 μM SB-431542 to inhibit signaling activity of endogenous receptors.

The online version of this article includes the following figure supplement(s) for figure 1:

**Figure supplement 1.** Luciferase reporter assay displaying activin C (ActC) titration against constant activin A (ActA) (0.62 nM) in untransfected (CAGA)$_{12}$-luciferase (luc).

functioning similarly to TGFβ1, suggesting an agonistic role of the ActC ligand (*Huang et al., 2020*; *Liu et al., 2012*).

TGFβ ligands are processed from precursor proteins comprises a prodomain, which aids in proper folding and ligand maturation, and a C-terminal signaling domain, which forms the covalent dimers (*Figure 1A*). The latter assemble receptor complexes on the cell surface containing a symmetrical positioning of two type I and two type II serine-threonine kinase receptors, which results in the activation of a SMAD signaling cascade (*Figure 1B*). There are seven type I receptors in the family termed activin receptor-like kinases 1 through 7 (ALK1-7) (*Derynck and Budi, 2019*). For ligands of the activin class, the type II receptors bind with high affinity (nM) to each individual chain in the dimer, with the low-affinity type I receptors binding at a composite interface formed by the two dimer chains (*Figure 1A*; *Goebel et al., 2019a*; *Goebel et al., 2019b*).

The activins, as a class, bind to and signal through three type I receptors: ALK4, ALK5, and ALK7. In general, each member signals through ALK4, whereas GDF8 and GDF11 extend specificity to ALK5 and ActB to ALK7 (*Hinck et al., 2016*; *Walker et al., 2017*). Structural and biochemical studies have made strides in illuminating the determinants of specificity between the activins and the type I receptors, providing context for the different biological roles of each activin member (*Goebel et al., 2019a*; *Goebel et al., 2022*; *Sako et al., 2010*; *Groppe et al., 2008*). ALK4 and ALK5 expression is relatively widespread, while ALK7 is primarily expressed in the adipose and reproductive tissues, and also in the brain and pancreatic cells (*Bondestam et al., 2001*; *Carlsson et al., 2009*). While ALK7-specific signaling has been linked to cancer cell apoptosis, its role in adipose tissue is more well-studied (*Michael et al., 2019*; *Li and Ventura, 2019*). ActB signaling via ALK7 in adipose tissue suppresses lipolysis and downregulates adrenergic receptors, facilitating fat accumulation (*Andersson et al., 2008*; *Yogosawa et al., 2013*; *Guo et al., 2014*). Similarly, loss of signaling in ALK7 knockout mice renders the animals resistant to diet-induced obesity (*Andersson et al., 2008*; *Yogosawa et al., 2013*; *Guo et al., 2014*).

Unlike the other activin ligands, limited information is available for the signaling capacity of ActC and whether the homodimer can actually activate SMAD molecules. One hypothesis is that ActC is a non-signaling molecule and simply a non-functional byproduct of expressing InhβC in the presence of InhβA. Given this uncertainty, we sought to characterize the signaling capacity of ActC across the panel TGFβ type I receptors. We demonstrate that homodimeric ActC can act as a potent activator of SMAD2/3 and does so with high specificity via the type I receptor, ALK7. Additionally, unlike the rest of the activin class, ActC has a much lower affinity for the type II receptors, ActRIIA and ActRIIB. Intriguingly, ActC is not antagonized by follistatin, which potently neutralizes ActA, ActB, GDF8, and GDF11. Finally, we demonstrate that ActC can activate SMAD2/3 signaling similarly to ActB in mature adipocytes in an ALK7-dependent manner.

## Results

### Activin C induces SMAD2/3 phosphorylation through ALK7

Cell-based reporter assays have long been used to measure SMAD activation. To investigate ActC's ability to induce canonical SMAD2/3 signaling like other activins, we performed luciferase reporter assays in an activin-responsive HEK293T cell line stably transfected with $(CAGA)_{12}$-luciferase (luc) plasmid (*Walker et al., 2017*; *Goebel et al., 2022*). Purified recombinant activin ligands (ActA, ActAC, ActC, and ActB) were titrated to generate EC50 curves. In this format, ActA stimulated a response at a lower ligand concentration than either ActAC or ActB (*Figure 1C*, left panel). In contrast, ActC did not induce reporter activity up to concentrations of 5 nM. Of note, ActAC showed about half of the activity of ActA, consistent with ActAC being less potent than ActA but more potent than ActC. Neither ActAC nor ActC activated a SMAD1/5/8-dependent reporter in an osteoblast cell line, in contrast to the robust response observed with BMP2 (*Figure 1C*, right panel).

Though the above data show that ActC does not signal like other activin ligands, HEK293 cells endogenously express only two of the three SMAD2/3 type I receptors, ALK4, and ALK5, with little to no expression of ALK7 (*Walker et al., 2017*; *Bu et al., 2018*). To address this limitation, we applied a heterologous system developed to interrogate specific signaling from individual type I receptors (*Goebel et al., 2019a*). Here, a point mutation was introduced into each type I receptor (ALK4 S282T [ALK4$_{st}$], ALK5 S278T [ALK5$_{st}$], or ALK7 S270T [ALK7$_{st}$]) that rendered it resistant to the

small molecule kinase inhibitor, SB-431542, while maintaining ligand-induced activation. HEK293T (CAGA)$_{12}$-luc reporter cells were transiently transfected with the modified receptors, then co-treated with the indicated ligands and SB-431542 to suppress signaling from endogenous receptors. In the presence of ALK4$_{st}$, ActA, ActAC, and ActB, but not ActC, stimulated reporter activity (*Figure 1D*). In ALK5$_{st}$-transfected cells, none of the activins stimulated reporter activity, while GDF11, a known ALK5 ligand, served as a positive control (*Figure 1D*). As expected, ActB and, to a much lesser extent, ActA induced reporter activation when ALK7$_{st}$ was transfected into the HEK293T (CAGA)$_{12}$-luc reporter cells. Strikingly and unexpectedly, ActAC and ActC activated (CAGA)$_{12}$-luc activity in the presence of ALK7$_{st}$ to a similar extent as ActB (*Figure 1D*).

These data suggested that ActC can signal specifically through ALK7. To further validate these results, we utilized an antibody that was developed to specifically bind and neutralize ligand signaling through ALK7. Following treatment with the anti-ALK7 antibody, ActB signaling via ALK7st was significantly reduced while ActC signaling was nearly abrogated completely (*Figure 1E*). The specificity of the antibody for ALK7 was confirmed as ActA signaling via ALK4$_{st}$ was unaffected (*Figure 1E*). Taken together, these data show that the activin ligands have differential type I specificities. ActA signals predominantly through ALK4; ActB and ActAC signal through ALK4 and ALK7; while ActC signals exclusively through ALK7.

Since ActC does not signal through ALK4, we wanted to test whether or not ActC could act as an antagonist of ActA signaling, as previously suggested (*Gold et al., 2009*). Along these lines, titration

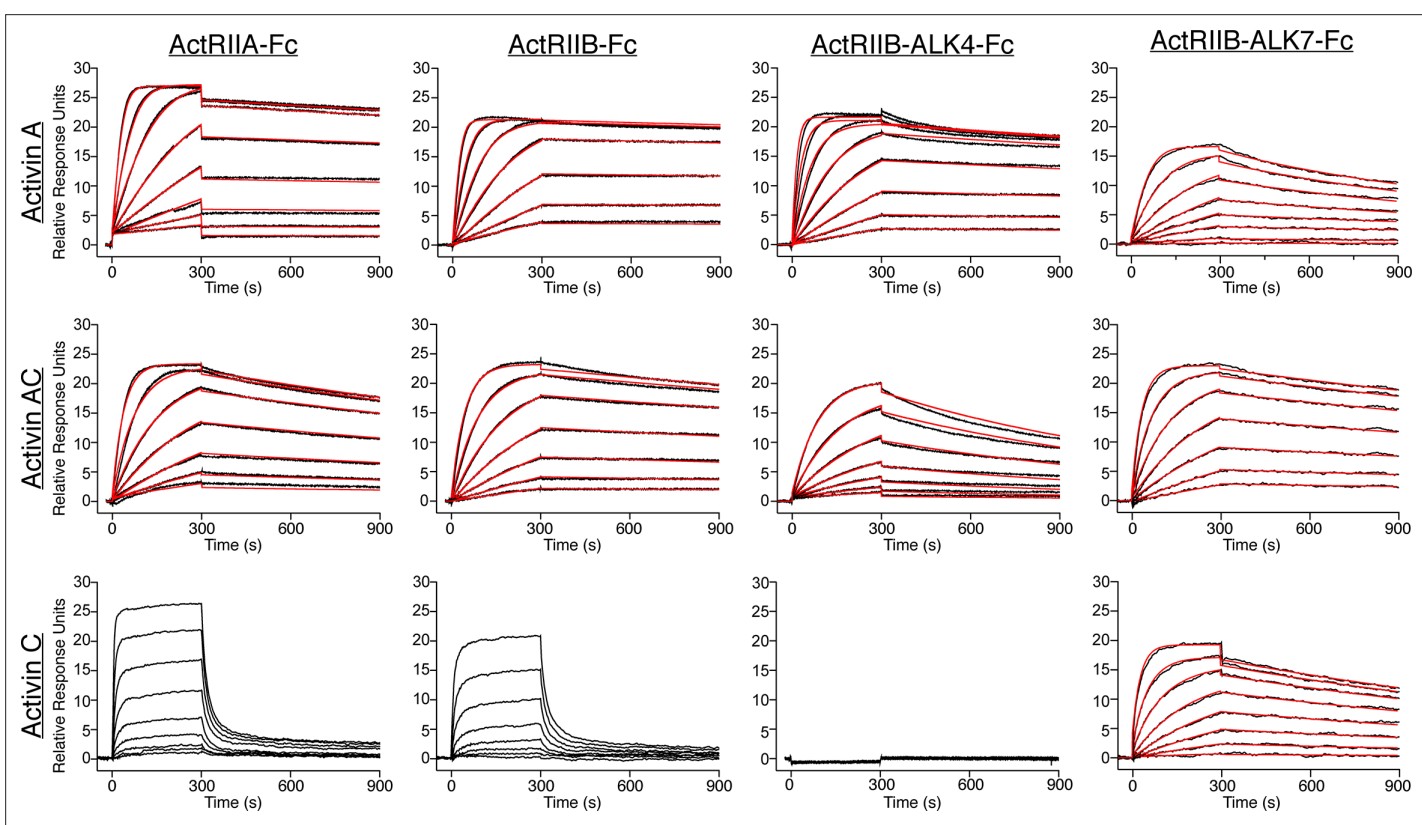

**Figure 2.** ActC binds activin type II receptors with low affinity. Representative surface plasmon resonance (SPR) sensorgrams of activin A (ActA), activin AC (ActAC), and activin C (ActC) binding to protein A captured ActRIIA-Fc, ActRIIB-Fc, ActRIIB-ALK7-Fc, or ActRIIB-ALK4-Fc. Sensorgrams (black lines) are overlaid with fits to a 1:1 binding model with mass transport limitations (red lines). ActC binding to ActRIIA and ActRIIB were fit using a steady state model. Each experiment was performed in duplicate with the kinetic parameters summarized in *Supplementary file 2*.

The online version of this article includes the following figure supplement(s) for figure 2:

**Figure supplement 1.** Additional surface plasmon resonance (SPR) sensorgrams of ActA, ActAC, and ActC binding to Alk7 and ActB binding to different receptors.

**Figure supplement 2.** Native gel analysis of type II receptors and ActA, ActAC, and ActC.

of ActC against constant ActA resulted in no significant reduction in signaling (*Figure 1—figure supplement 1*). These results show that extracellular ActC does not antagonize ActA signaling.

## Activin C and activin AC interact with and require activin type II receptors to signal

In addition to the type I receptors, TGFβ family ligands must bind type II receptors to generate intracellular signals. Ligands of the activin class generally bind the type II receptors ActRIIA and ActRIIB with high affinity (pM-nM) (*Goebel et al., 2019a*; *Sako et al., 2010*; *Pearsall et al., 2008*; *Oh et al., 2002*; *Harrison et al., 2006*). Previous studies showed that ActAC binds ActRIIB with lower affinity than ActA, suggesting that the InhβC chain has a diminished type II interaction (*Gold et al., 2013*). Given the new finding that ActC is a signaling molecule, we sought to determine and compare the binding affinities of ActA, ActAC, and ActC to ActRIIA, and ActRIIB using surface plasmon resonance (SPR). In this experiment, type II receptor extracellular domains (ECDs) fused to an antibody Fc fragment were captured using a protein A biosensor chip, while ActA, ActAC, or ActC were titrated as the analyte. For both ActRIIA-Fc and ActRIIB-Fc, binding affinity was highest for ActA (equilibrium constant [apparent $K_D$] of 22 pM and 8.1 pM, respectively) (*Figure 2*, top row). ActAC also bound ActRIIA and ActRIIB with high affinity, although slightly weaker than ActA (150 pM and 90 pM, respectively), suggesting that the InhβC chain diminishes the overall type II affinity of the dimer (*Figure 2*, middle row). ActC binding to ActRIIA and ActRIIB was much weaker, with a significantly faster dissociation rate than either ActA or ActAC (*Figure 2*, bottom row). While ActA had no apparent preference for one type II receptor, consistent with previous studies, ActC had a higher affinity for ActRIIA than ActRIIB (*Figure 2—figure supplement 1*; *Goebel et al., 2019a*). Similarly, ActB binding to type II receptors was much stronger then ActC (*Figure 2—figure supplement 1B*), indicating that ActC deviates from other activin class ligands by exhibiting low affinity for type II receptors. To confirm the weak binding of ActC towards type II receptors, we performed a native gel analysis, where ActA, ActAC, and ActC were incubated with either ActRIIA or ActRIIB. Both ActA and ActAC, but not ActC, formed a stable complex when incubated with either ActRIIA or ActRIIB (*Figure 2—figure supplement 2*). Collectively, these data indicate that ActC deviates from other activin class ligands by exhibiting low affinity for type II receptors.

We also used SPR to determine if ActC or ActAC bind specifically to ALK7. No to little binding of any of the activins, including ActC and ActAC, was observed for ALK7-Fc, indicating that for all ligands ALK7 is a low-affinity receptor (*Figure 2—figure supplement 1C*). With a low-affinity type II and type I receptor, we asked whether the combination of receptors enhanced binding of ActC. A heterodimeric-Fc receptor fusion that incorporates both the type I and type II receptor can mimic natural signaling pairs (*Li et al., 2021*). Previously, ActRIIB-ALK4-Fc exhibited higher affinity for ActA than the monovalent ActRIIB-Fc, indicating enhanced binding due to incorporation of the type I receptor (*Goebel et al., 2022*). We therefore tested binding of the heterodimeric ActRIIB-ALK7-Fc to ActA, ActAC, and ActC (*Figure 2*). ActA bound ActRIIB-ALK7-Fc (296 pM) with sevenfold lower affinity than ActRIIB-Fc (42 pM), indicating ALK7 did not contribute to binding, consistent with ActA not signaling via ALK7. Interestingly, significant binding was observed for both ActC (2 nM) and ActAC (51 pM) to ActRIIB-ALK7. While for ActAC, binding to ActRIIB-ALK7-Fc was slightly higher than binding to ActRIIB-Fc (64 pM), a dramatic difference was observed for ActC where binding was increased 40-fold over ActRIIB-Fc alone. Similar studies were performed with ActRIIB-ALK4-Fc. As expected ActA bound with high affinity to ActRIIB-ALK4-Fc while ActAC had a much weaker interaction (457 pM), and ActC failed to bind (*Figure 2*). SPR experiments with ActB showed consistent results, where high-affinity interactions were observed with ActRIIA, ActRIIB, and ActRIIB-ALK7-Fc, with little to no binding to ALK7-Fc (*Figure 2—figure supplement 1C*). Binding data for each SPR experiment can be found in *Supplementary file 2*.

Next, we investigated whether ActC signaling could be inhibited using the type II receptor-Fc constructs as a competitive antagonist (ligand trap) to block endogenous receptor binding in a cell-based assay (*Figure 3A*). We employed the same assay system in which SB-431542 resistant type I receptors were transfected into HEK293T (CAGA)$_{12}$-luc reporter cells. We titrated either ActRIIA-Fc or ActRIIB-Fc against a constant concentration (0.62 nM) of ActA, ActAC, ActC, or ActB (*Figure 3B and C*). ActRIIA-Fc and ActRIIB-Fc dose-dependently inhibited ActA and ActAC signaling via ALK4$_{st}$ and ablated ActB signaling via ALK7$_{st}$. In contrast, signaling by both ActAC and ActC through ALK7$_{st}$ was

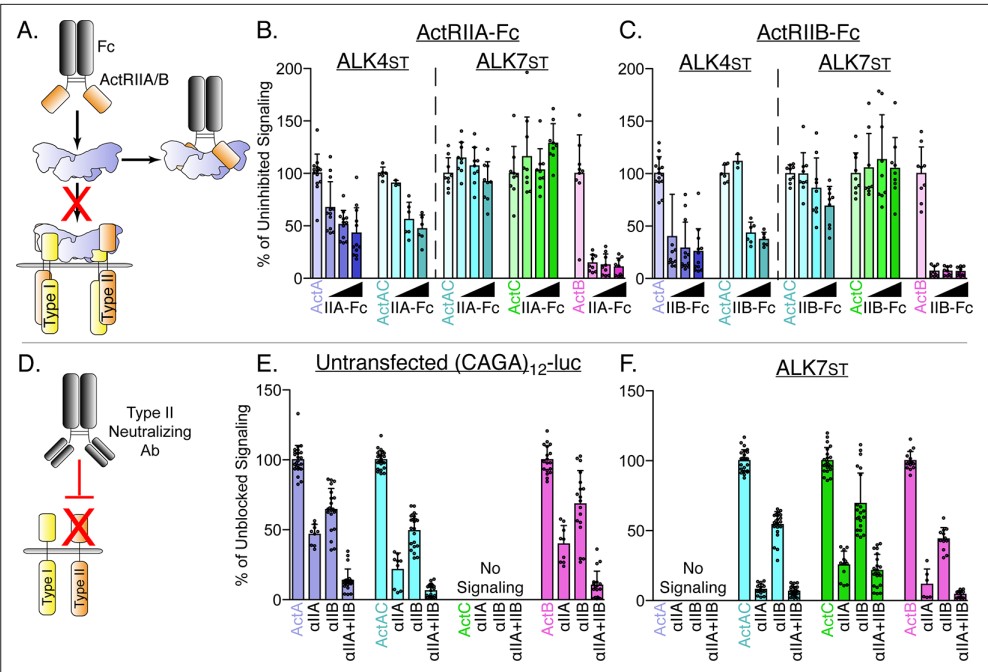

**Figure 3.** ActA and ActAC signal via activin type II receptors. (**A**) Schematic representation of activin type II receptor Fc-fusion proteins as decoys. (**B and C**) HEK293T (CAGA)$_{12}$-luciferase (luc) cells were transfected with ALK4$_{st}$ and ALK7$_{st}$ and treated with SB-431542 and activin A (ActA), activin AC (ActAC), activin (ActC), or activin B (ActB) (0.62 nM) as in *Figure 1* in the presence of increasing quantities of either ActRIIA-Fc (**B**) or ActRIIB-Fc (**C**). (**D**) Schematic representation of neutralizing antibodies targeting the type II receptor extracellular domains (ECDs). (**E and F**) HEK293T (CAGA)$_{12}$-luc cells following treatment with ActA, ActAC, ActC, or ActB (0.62 nM) in the presence or absence of neutralizing antibodies targeting ActRIIA, ActRIIB, or both. No signaling was observed by ActC in (E*)* or ActA in (F*)*. Each data point represents technical replicates within triplicate experiments measuring relative luminescence units (RLU) with bars displaying the mean ± SD. Data are represented as percentage of uninhibited (**B and C**) or unblocked (**E and F**) signal.

not inhibited by ActRIIA-Fc or ActRIIB-Fc, even at the highest concentration of the decoy receptors (25 nM).

Given its low affinity for activin type II receptors, we wanted to determine whether ActC signaling through ALK7 was dependent on ActRIIA or ActRIIB. We therefore used a series of receptor neutralizing antibodies, which bind to either the ECD of ActRIIA or ActRIIB, or to both receptors (*Figure 3D*). As expected, neutralization of either ActRIIA or ActRIIB significantly reduced signaling by ActA and ActB in the untransfected and untreated (i.e. without SB-431542) CAGA-luc cells (*Figure 3E*). An antibody that simultaneously blocks both ActRIIA and ActRIIB more potently inhibited signaling of each activin ligand than the single-target antibodies, indicating that ActRIIA and ActRIIB were redundant. ActAC signaling in the untransfected/untreated CAGA-luc cells was also inhibited by type II receptor neutralization in a similar manner to ActA and ActB. Again, ActC did not signal under these assay conditions unless ALK7$_{st}$ was added. Here, both ActAC and ActB signaling was readily inhibited by type II receptor blockade (*Figure 3F*). ActC signaling was significantly reduced when ActRIIA was blocked and to a lesser extent with blocking ActRIIB. These observations demonstrate that, despite their lower affinities, ActC and ActAC require a type II receptor, with a preference for ActRIIA, for signaling via ALK7.

## Activin C is antagonized by inhibin A, but not follistatin-288 or follistatin-like protein 3

Activin class ligands are regulated through several mechanisms. One such mechanism is through extracellular antagonists, such as follistatin-288 (Fst-288) and follistatin-like 3 (FSTL3), which bind and sequester ligands. Fst-288 and FSTL3 form a donut-like shape to surround activin ligands, occluding epitopes that are important for binding to both type I and type II receptors (*Cash et al., 2009*; *Cash*

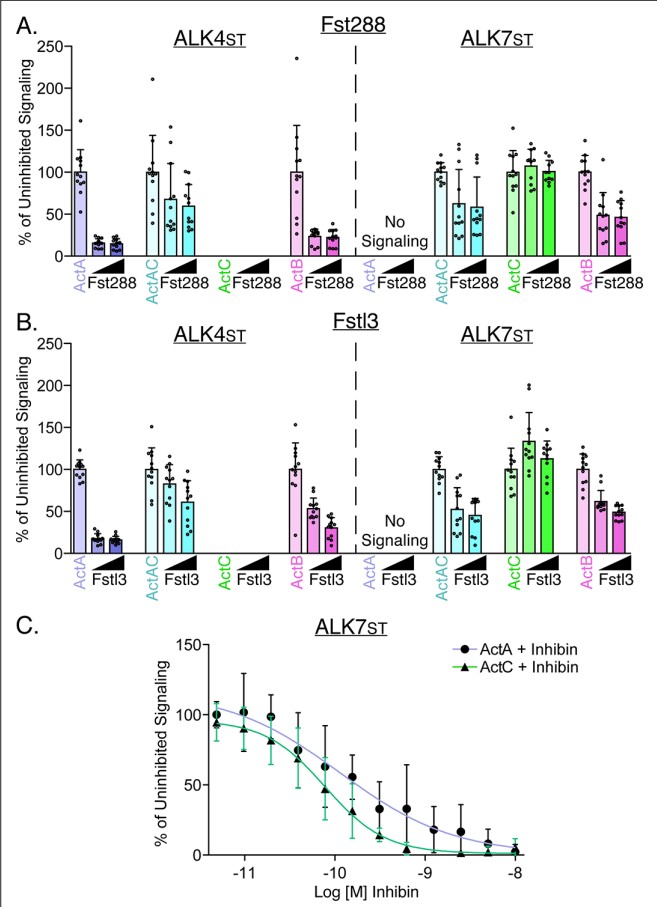

**Figure 4.** Activin C is resistant to inhibition by follistatin but not inhibin A. (**A**) HEK293T (CAGA)₁₂-luciferase (luc) cells were transfected with ALK4st and ALK7st and treated with SB-431542 and activin A (ActA), activin AC (ActAC), activin C (ActC), or activin B (ActB) (0.62 nM) with increasing quantities (12.5 nM or 25 nM) of either follistatin-288 (Fst-288) (**A**) or follistatin-like 3 (Fstl3) (**B**). (**C**) Luciferase assay following treatment with either ActA (ALK4st signaling) or ActC (ALK7st signaling) at a constant concentration (0.62 nM) along with titration of inhibin A (InhA). Each data point represents technical replicates within triplicate experiments measuring relative luminescence units (RLU) with bars displaying the mean ± SD. Data are represented as percentage of uninhibited. IC50 values are reported in **Supplementary file 1**.

et al., 2012a; **Cash et al., 2012b**; **Thompson et al., 2005**; **Stamler et al., 2008**). Given the new finding that ActC can signal via type I (ALK7) and II (ActRIIA/B) receptors, we next examined whether its actions were inhibited by either Fst-288 or FSTL3. Fst-288, at two concentrations, robustly inhibited ActA and ActB induction of CAGA-luc activity via ALK4st (**Figure 4A**). Fst-288 also inhibited ActB signaling via ALK7st, though to a lesser extent. Fst-288 moderately inhibited ActAC signaling via ALK4st and ALK7st, but unexpectedly had no impact on ActC actions (**Figure 4A**). Similar results were observed with the related antagonist FSTL3, which binds and occludes activin ligands similarly to Fst-288 (**Figure 4B**; **Stamler et al., 2008**; **Sidis et al., 2006**). This unique resistance to classical activin antagonists further distinguishes ActC from the rest of the activin class.

Inhibins are ligand-like antagonists of activins that are formed from the heterodimerization of an InhβA or InhβB chain and the Inhα chain, resulting in the heterodimers inhibin A and B (**Bernard et al., 2001**). These heterodimers act as a signaling dead molecules by binding type II receptors in a non-productive receptor complex. To test whether inhibin A can antagonize ActC signaling, we titrated recombinant inhibin A against ActC in the above-described ALK7st luciferase assay. Like ActA, ActC signaling was dose-dependently attenuated by InhA with an IC50 value of 0.08 nM (SD ± 0.04 nM) as compared to 0.2 nM for ActA (SD ± 0.2 nM) (**Figure 4C**).

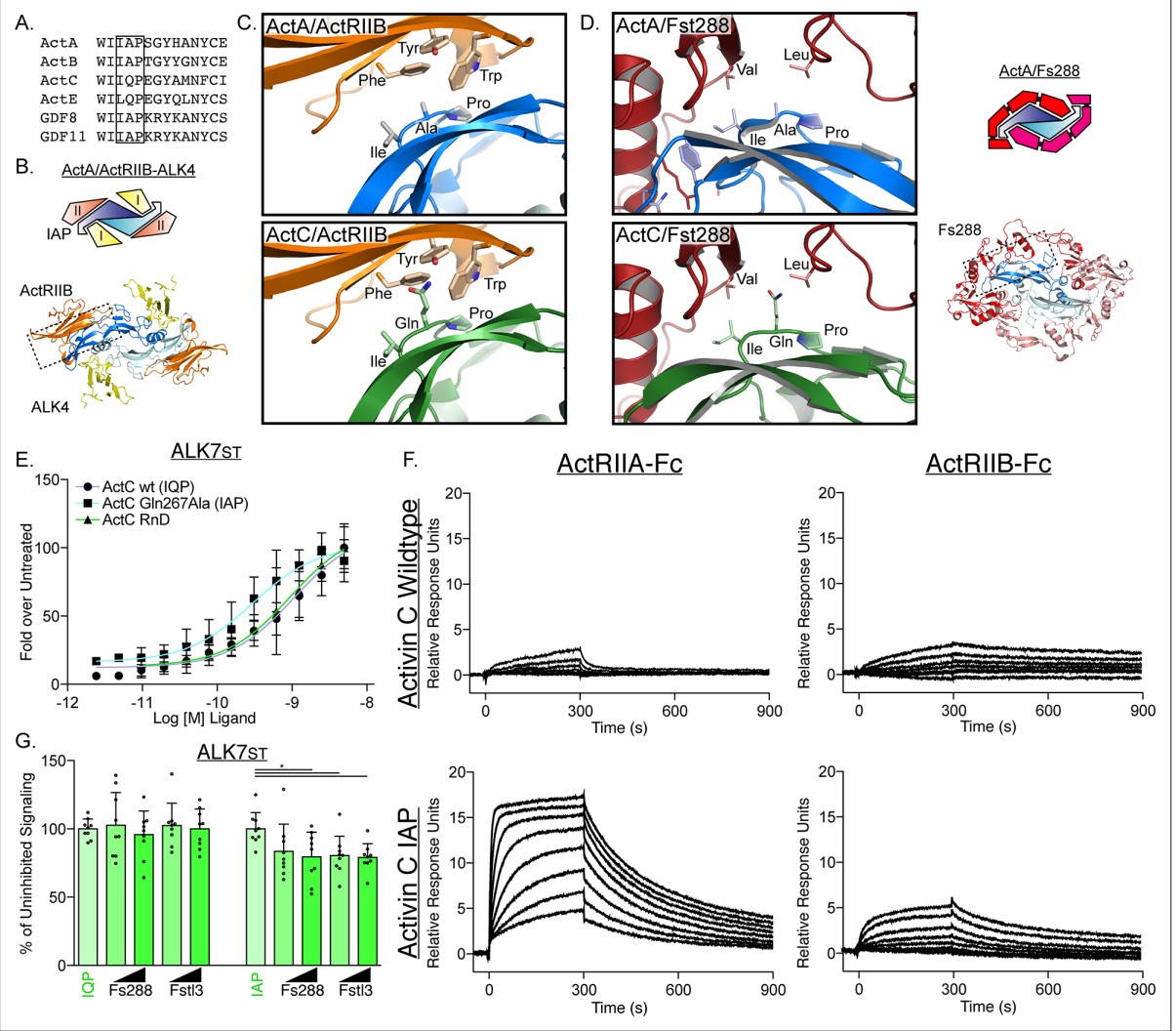

**Figure 5.** Type II interface of ActC is distinct from other activins. (**A**) Sequence alignment across the activin class displays critical differences at the canonical type II receptor binding site. IAP motif is boxed in black. (**B**) Structure and schematic representation of the ActA/ActRIIB/ALK4 complex (PDB: 7OLY). ALK4 is in yellow, ActRIIB is in orange, activin A (ActA) is in blue. (**C**) Comparison of type II receptor interface between ActA/ActRIIB and the ActC/ActRIIB model, centered on the IAP (ActA) and IQP (activin C [ActC]) motifs. The ActC model was built from (PDB:7OLY) (*Thompson et al., 2005*). (**D**) Comparison (left) of the Fst-288 interface between ActA/Fst-288 (PDB:2BOU) and the ActC/Fst-288 model, centered on the IAP (ActA) and IQP (ActC) motifs. Schematic of ActA/Fst-288 included (right) for clarification. (**E**) ALK7$_{st}$-dependent luciferase assay following treatment with ActC purchased from R&D systems, or recombinant ActC wildtype (WT) (IQP) or ActC Gln267Ala (IAP) transiently produced in HEK293T cells. Each data point represents the mean ± SD of triplicate experiments measuring relative luminescence units (RLU). EC50 values are reported in *Supplementary file 1*. (**F**) Average surface plasmon resonance (SPR) sensorgrams of ActC WT (IQP) and ActC Gln267Ala (IAP) binding to protein A captured ActRIIA-Fc or ActRIIB-Fc. Sensorgrams (black lines) are overlaid with fits to a 1:1 binding model with mass transport limitations (red lines). Each experiment was performed in duplicate. (**G**) ALK7$_{st}$-dependent luciferase reporter assay following treatment of ActC WT (IQP) and ActC Gln267Ala (IAP) (0.62 nM) with increasing concentrations (12.5 nM or 25 nM) of either Fst-288 or Fstl3. Each data point represents technical replicates within triplicate experiments measuring relative luminescence units (RLU) with bars displaying the mean ± SD.

## Modeling of the activin C ligand

Given the low affinity of ActC for the activin type II receptors and the ligand's resistance to follistatin inhibition, we next examined what molecular differences within the activin class ligands could account for variation in ligand-receptor or ligand-follistatin interactions. A trio of residues (Ile340, Ala341, and Pro342; IAP motif; ActA) at the ligand knuckle were utilized during both type II receptor and follistatin binding (*Harrison et al., 2006*; *Thompson et al., 2005*; *Stamler et al., 2008*; *Thompson et al., 2003*). Sequence alignment across the activin class reveals conservation of this motif in each ligand of the activin family, except for ActC and ActE (*Figure 5A*). During complex formation between

ActA:ActRIIB, the IAP motif forms the core of interactions with a series of hydrophobic residues in ActRIIB (Tyr60, Trp78, Phe101) (*Figure 5B and C*). Additionally, this interface is engaged by Fst-288, highlighting that the IAP motif is utilized by both antagonists and receptors (*Figure 5D*). The core interactions involving the IAP motif are consistent across other structures within the activin class, such as GDF8:Fst-288, GDF11:ActRIIB, and GDF11:Fst-288 (*Goebel et al., 2019b*; *Walker et al., 2017*). In comparison, ActC contains a glutamine residue in place of the central alanine residue of the IAP motif. Generating a model of ActC (swissmodel to ActA; PDB: 7OLY) and aligning it to ActA reveals that Gln267 of ActC would be sterically unfavorable for interactions with both ActRIIB and Fst-288 (*Figure 5C, D*). Thus, we hypothesized that Gln267 in ActC might contribute to the reduced interaction with both the type II receptors and Fst-288.

To test this idea, we expressed and purified both recombinant wildtype (WT) (IQP) and Q267A (IAP) ActC from HEK293T cells. The IAP mutant had similar activity to the IQP WT form of ActC in the ALK7$_{st}$-dependent CAGA-luciferase assay (*Figure 5E*). As determined by SPR, ActC WT had low affinity for both ActRIIA-Fc and ActRIIB-Fc (*Figure 5F*), consistent with binding data using recombinant ActC from R&D systems. Replacement of the glutamine with alanine in ActC resulted in an increase in type II receptor binding, especially for ActRIIA-Fc (620pM) (*Figure 5F* and *Supplementary file 2*). Next, we challenged ActC$^{Q267A}$ with follistatin in the CAGA-luc assay. Here, ActC$^{Q267A}$ was more inhibited by both Fst-288 and Fstl3 than WT ActC (*Figure 5G*). Taken together, these data support the hypothesis that the glutamine substitution in ActC relative to the other ligands of the activin class (ActA, ActB, GDF8 and GDF11) weakens the affinity for both the type II receptors and follistatin.

## Activin C signals similarly to activin B in mature adipocytes

Since we have shown that ActC can activate ALK7 in a cell-based luciferase assay, we next sought to determine the ligand's capacity to signal via endogenous ALK7 in a biologically relevant cell type, adipocytes. To this end, we utilized both the preadipocyte cell line, 3T3-L1, and mature adipocytes derived from the stromal vascular fraction (SVF) of murine adipose tissue. Cells were differentiated over 4 days and maintained for 6 further days, where cell morphology and lipid droplets visibly increased, indicative of mature adipocytes (*Figure 6A*). ActC stimulated SMAD2 phosphorylation (pSMAD2) in adipocytes differentiated from SVF but not 3T3-L1 cells (*Figure 6B, C*). Notably, ALK7 (product of the *Acvr1c* gene) expression was significantly higher in SVF relative to 3T3-L1-derived adipocytes (*Figure 6D*). ActB stimulated pSMAD2 in both cell types, presumably via ALK4 in 3T3-L1 or a combination of ALK4 and ALK7 in differentiated adipocytes (*Figure 6B, C*). In the mature (SVF-derived) adipocytes, both ActB and ActC induced pSMAD2 in a similar manner; however, Fst-288 only blocked ActB action (*Figure 6C*), consistent with the results above in the ALK7$_{st}$-dependent luciferase assay (*Figure 4A*). The neutralizing ALK7 antibody blocked ActC-induced pSMAD2 in mature adipocytes supporting that signaling was dependent on the ALK7 receptor (*Figure 6E*). Interestingly, ActB induced pSMAD2 was only partly blocked in this assay, likely due to residual signaling via ALK4 (*Figure 6E*). These results demonstrate that ActC is an ALK7-dependent signaling ligand and is follistatin resistant in a physiologically relevant context (*Figure 6C and E*).

ActB has dual effects on adipogenesis, and its function depends on the relative expression of ALK4 and ALK7 during the process of adipocyte commitment and differentiation (*Carlsson et al., 2009*; *Guo et al., 2014*; *Ibáñez, 2021*; *Kogame et al., 2006*). ActA or ActB exposure during differentiation of SVF cells, when ALK7 levels are low, inhibits adipogenesis (*Figure 6F*). Treatment with ActC at this early stage did not affect adipogenesis (*Figure 6F*). Follistatin antagonized the anti-adipogenic effects of both ActA and ActB, restoring normal adipogenesis and lipid droplet formation (*Figure 6F*). Furthermore, gene expression of both *Pparg2* and *Cebpa*, essential transcription factors for adipogenesis, was impaired by ActA or ActB but not ActC (*Figure 6G*). However, ActC significantly reduced both *Pnpla2* expression and lipid content, consistent with the late-stage, proadipogenic effects of ActB-ALK7 signaling (*Figure 6G*; *Ibáñez, 2021*; *Hoggard et al., 2009*; *Hirai et al., 2005*; *Nielsen, 2009*).

## Discussion

The binding/signaling profiles of some members of the activin class (ActA, ActB, GDF8, and GDF11) of TGFβ ligands have been largely characterized, where each member exhibits differential specificity

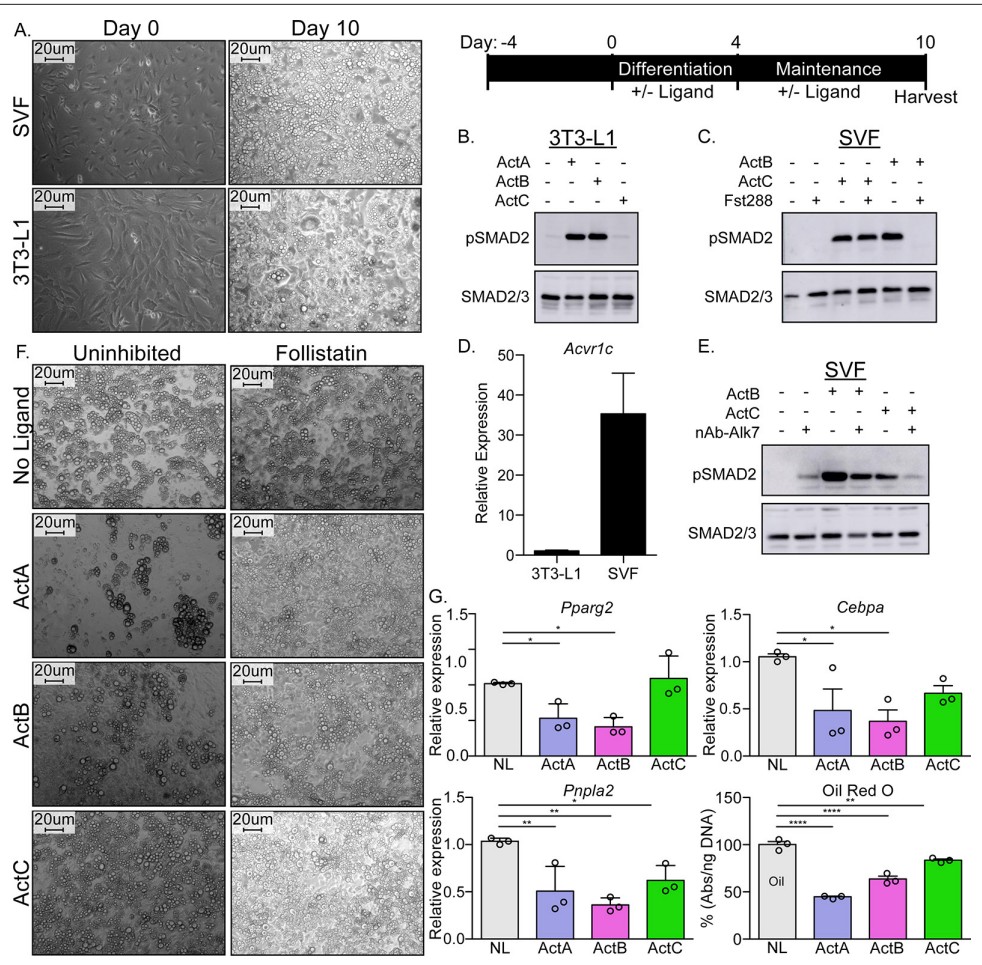

**Figure 6.** ActC activates SMAD2 through ALK7 in differentiated adipocytes. (**A**) Representative images of isolated adipose-derived stromal vascular fraction (SVF) or cultured 3T3-L1 cells prior to differentiation (left, Day-0) and following differentiation (right, Day-10). Scale bars are 20 µm. Schematic shown in upper right for visualization of timeline. (**B**) Western blot (WB) showing phosphorylated SMAD2 (pSMAD2) and total SMAD2/3 in 3T3-L1-derived adipocytes following treatment with activin A (ActA), activin B (ActB), or activin C (ActC) (2 nM) for 1 hr. (**C**) WB following treatment of SVF-derived adipocytes with ActB or ActC (2 nM) with or without Fst-288 (800 ng/ml) for 1 hr. (**D**) Quantitative PCR (RT-qPCR) of *Acvr1c* expression in differentiated 3T3-L1 cells and SVF adipocytes. Bars display mean ± SD of three experimental replicates. (**E**) WB following treatment of SVF-derived adipocytes with ActB or ActC (2 nM) in the presence or absence of a neutralizing antibody targeting ALK7 for 1 hr. (**F**) Representative images of SVF-derived adipocytes following treatment with ActA, ActB, or ActC during differentiation with or without Fst-288. (**G**) RT-qPCR of target genes *Pparg2*, *Cebpa*, and *Pnpla2* following treatment with ActA, ActB, or ActC during differentiation. Oil Red O quantification based on images in (**F**). Significance is represented as: * p<0.05, ** is p<0.01, *** p<0.001 and **** p<0.0001. Each experiment was performed in triplicate. While representative westerns are shown, supplemental westerns can be found in *Figure 6—figure supplement 1*.

The online version of this article includes the following figure supplement(s) for figure 6:

**Figure supplement 1.** Supplemental adipocyte-pSMAD2/SMAD2/3 western blots.

for both the type II receptors, ActRIIA and ActRIIB, and the type I receptors: ALK4, ALK5, and ALK7. ActA is limited to a single type I receptor, ALK4, and has little type II receptor preference, while GDF11 can promiscuously signal through ALK4, ALK5, and ALK7 and seemingly favors interaction with ActRIIB (*Goebel et al., 2019a*). The receptors for ActC have remained largely unknown, in part due to the initial characterization of ActC as a non-signaling molecule (*Butler et al., 2005*). In this study, we identified ActC as a bona fide signaling ligand with distinct molecular properties from other activin class ligands.

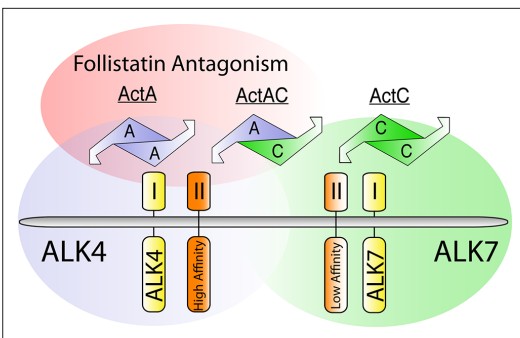

**Figure 7.** Difference between ActA and ActC in type I receptor specificity, type II receptor affinity, and follistatin antagonism. Gradients of follistatin antagonism (red), activin receptor-like kinase 4 (ALK4)-dependent signaling (blue), activin receptor-like kinase 7 (ALK7)-dependent signaling (green), and activin type II receptor affinity (orange) for activin A (ActA), activin AC (ActAC), and activin C (ActC). Ligands and type I receptors are shown schematically.

The online version of this article includes the following figure supplement(s) for figure 7:

**Figure supplement 1.** Sequence alignment of activin and TGFβ ligands.

**Figure supplement 2.** Modeling ActC/ALK7 interactions.

**Figure supplement 3.** Phylogenetic history of ActC.

ActC signals through ALK7, whereas ActAC uses both ALK4 and ALK7 (*Figure 7*). Thus, ligands that contain an InhβC subunit can bind and act through ALK7. This is similar to what was previously described for ligands containing an InhβB subunit, like ActB and ActAB. In contrast, ActA does not signal through ALK7. Interestingly, the heterodimer ActAC was more potent than ActC, which has similarly been reported for other heterodimers in the family, such as BMP2/4 and BMP2/7 (*Kaito et al., 2018*; *Aono et al., 1995*). This might be a result of different type I receptor binding epitopes that are formed in the heterodimer versus the homodimer.

The molecular basis for type I receptor specificity remains an intriguing aspect of the evolution of the activin class ligands. Initial studies implicated the wrist region, including the prehelix loop, as a major contributor towards type I receptor specificity, as swapping this region could alter type I receptor specificity (*Cash et al., 2009*). More recent studies have identified residues in the fingertip region of the ligand as also have a major role in type I receptor specificity (*Goebel et al., 2019a*; *Aykul et al., 2020*). Given the low-affinity nature of the type I receptors for ligands across the activin family, the current thought is that subtle differences at the type I: ligand interface dictate receptor specificity. Interestingly, fingertip residues that are important for ActA and GDF11 binding to type I receptors are divergent in ActC (*Figure 7—figure supplement 1*) and could account for the latter's lack of signaling through ALK4. Most notably, a recent crystal structure of ActA in complex with ALK4 shows that D406 of ActA forms a hydrogen bond with the mainchain of ALK4 (*Figure 7—figure supplement 2*). The corresponding residue is an arginine in ActC (*Goebel et al., 2022*). On the receptor side, the β4-β5 loop is important for ligand recognition and is three residues shorter in ALK7 than in ALK4 and ALK5, possibly to accommodate the larger arginine residue, which, in models, extends toward this loop (*Figure 7—figure supplement 2*). Certainly, structures of ALK7 in complex with ActC or other ligands will help determine how specificity for ALK7 is acquired. Regardless, it seems that differences in the ligand fingertip and/or prehelix loop, coupled with differences in the receptor β4-β5 loop, dictate type I receptor specificity in the activin class.

In general, activin class ligands bind activin type II receptors with high affinity (*Goebel et al., 2019a*; *Goebel et al., 2022*; *Harrison et al., 2006*; *Thompson et al., 2003*). Conversely, ActC exhibits weak binding to both ActRIIA and ActRIIB. ActAC binds the type II receptors but with reduced affinity compared to ActA. Thus, the InhβC subunit appears to reduce affinity for the activin type II receptors. Nevertheless, the type II receptors are required for ActC signaling, as the ligand's activity was abrogated by activin type II receptor neutralizing antibodies and inhibin A. Unlike the other members of the activin class, ActC binds both type I and type II receptors with low affinity but is still able to signal. One possible explanation is avidity from binding contribution both the type I and type II receptors are needed for ActC to signal. This is supported by SPR studies with the heteroreceptor combination of ALK7-ActRIIB, which has a higher affinity for ActC than either receptor alone (*Figure 2*). It has been suggested that exogenous ActC can directly antagonize ActA signaling (*Gold et al., 2009*) whereby ActC might bind type II receptors but not signal functioning as a competitive inhibitor to ActA. Our data challenges this idea as exogenous ActC does not antagonize ActA (*Figure 1—figure supplement 1*), likely due to lack of ALK4 activity (*Figure 1*) and weak binding to ActRIIA, ActRIIB, and the fusion ActRIIB-ALK4-Fc (*Figure 2*). Thus, the main mechanism of ActC antagonism of ActA is likely

through heterodimer formation, where the ActAC ligand signals less potently through ALK4, while gaining the ability to signal through ALK7, when present.

Another unexpected characteristic of ActC is its interaction or lack thereof with the extracellular antagonists, the follistatins. Given ActC's structural similarity to other activin class members, it was unexpected that neither Fst-288 nor Fstl3 inhibited ActC. Similarly, suppression of ActAC signaling was less significant compared to the other activin ligands, ActA and ActB (*Figure 4*). This indicates that the InhβC chain limits follistatin binding and confers resistance to antagonism. The biological implications of follistatin resistance will need to be further explored, but it is tempting to speculate that the presence of follistatin would interfere with ligands that signal through ALK4, providing a permissive environment for the lower affinity ActC to bind type II receptors and signal via ALK7. Additionally, the presence of follistatin, while limiting ActA and ActB signaling, would still permit ActAC signaling via ALK4.

Having low affinity for the type II receptors and resistance to follistatin distinguishes ActC from the other members of the activin class and raises the question as to what confers differences in ligand properties, especially given their > 50% sequence identity. Comparison across the activins revealed conservation of the shared type II/follistatin binding surface except for a single residue, centrally located in the interface. While most activins have an alanine at this position, InhβC contains a glutamine akin to the glutamate within the TGF-β's (TGF-β 1–3) (*Figure 7—figure supplement 1*). Converting this residue to an alanine in ActC resulted in a ligand with a higher affinity for ActRIIA and increased sensitivity to follistatin. While not the only molecular difference, it appears that InhβC has evolved a single substitution centrally located in a major binding epitope compared to other activin class members that suppresses its interaction with follistatin and type II receptors. Comparison across different species shows this deviation is conserved in mammals (*Figure 7—figure supplement 3*). Interestingly, fish are divergent and possess the alanine version of InhβC similar to InhβA and InhβB. This bifurcation in conservation suggests differentially evolved activin ligands exist in the two taxa and might provide clues as to the biological function of ActC in different species.

Physiological roles for ActC and ActAC have not yet been established. However, given the ability of the ligands to signal via ALK7, we turned our attention to adipocytes. Human adipose tissue is a major site of both ActB and ALK7 expression, where the pair induces pro-obesity signaling outcomes, such as catecholamine resistance or inhibition of lipolysis (*Guo et al., 2014*; *Ibáñez, 2021*). In this study, we show that ActC regulates adipocyte differentiation differently than ActA or ActB. While both ActA and ActB exhibit potent inhibitory effects in cultured adipocytes, ActC does not. This difference is explained, in part, by the ability of ActA and ActB, but not ActC, to signal via ALK4, which is present in pre-adipocytes and throughout their maturation and differentiation. In contrast, ActC can signal in adipose tissue only once ALK7 is expressed, which occurs in the later stages of adipocyte differentiation. In this context, ActC negatively regulates lipase expression, lipid content, and elicits a SMAD2 response similar to that of ActB. In addition to ActB and ActC, the TGFβ ligand, GDF3, can also signal through ALK7. GDF3 signaling is supported by the co-receptor Cripto and is implicated in the regulation of energy homeostasis and adipocyte function (*Andersson et al., 2008*). Thus, taken together, it appears that several TGF-βligands have evolved the ability to signal in adipocytes depending on the receptor/co-receptor profile.

Given that ActC expression and secretion is highest within the liver, a tissue with low ALK7 expression, it is possible that ActC acts as a hepatokine functioning systemically in fat regulation (; *Schmitt et al., 1996*). Unlike ActB, ActC is not antagonized by follistatin and may therefore serve as an uninhibited, basal signal in the face of variable follistatin expression, such as during thermogenesis in adipocytes following cold exposure (*Braga et al., 2014*). Another possible role might be during liver regeneration, as ActAC and ActC have been observed in serum, coinciding with a surge of follistatin expression (*Gold et al., 2005*).

Another member of the activin class, ActE, is expressed highest in the liver and has been implicated as a hepatokine with a role in energy homeostasis; however, whether ActE can signal and, if so, through which receptors, has yet to be determined (*Sugiyama et al., 2018*). Interestingly, ActE has a glutamine in position 267, similar to ActC, and a leucine in position 265, replacing the conserved isoleucine (*Figure 7—figure supplement 1*). These amino acid differences likely reduce ActE's affinity for the type II receptors and follistatin, similar to ActC. Given these similarities, it is possible that there is functional overlap between ActC and ActE, possibly in the liver-adipose signaling axis.

Throughout the body, there are a variety of activins with differential receptor and antagonist binding, yielding a variety of potential signaling capacities (*Figure 7*). Our results indicate that ActC can act as a canonical TGFβ ligand, transducing SMAD2/3 responses similar to ActA and ActB while avoiding inhibition through the follistatin family of antagonists. This observation challenges current thinking that ActC acts solely as an activin antagonist. Different than ActB, which can signal through both ALK4 and ALK7, ActC is specific for ALK7 and shows a preference for the type II receptor ActRIIA. Future studies will need to address the different biological roles of ActC and ActAC signaling through ALK7, with an initial focus on adipose tissue.

# Materials and methods
## Protein expression and purification
### ActRIIA and ActRIIB
The extracellular domains of human ActRIIA (residues 1–134) and rat ActRIIB (residues 1–120) were produced as previously described (*Goebel et al., 2019a*). Specifically, both receptors were subcloned into the pVL1392 baculovirus vector with C-terminal Flag and $His_{10}$ tags (ActRIIA) or a C-terminal $His_6$ tag followed by a thrombin cleavage site (ActRIIB). Recombinant baculoviruses were generated through the Bac-to-Bac system (ActRIIA; Invitrogen - Waltham, MA) or the Baculogold system (ActRIIB; Pharmingen - San Diego,CA). Virus amplification and protein expression were carried out using standard protocols in SF + insect cells (Protein Sciences, Meriden, CT). ActRIIA and ActRIIB were purified from cell supernatants by using Ni Sepharose affinity resin (Cytiva, Marlborough, MA) with buffers containing 50 mM $Na_2HPO_4$, 500 mM NaCl, and 20 mM imidazole, pH 7.5 for loading/washing and 500 mM imidazole for elution. ActRIIB was digested with thrombin overnight to remove the $His_6$ tag. ActRIIA and ActRIIB were subjected to size exclusion chromatography (SEC) using a HiLoad Superdex S75 16/60 column (Cytiva) in 20 mM Hepes, and 500 mM NaCl, pH 7.5.

### ActRIIA-Fc, ActRIIB-Fc, ActRIIB-ALK7-Fc, ALK7-Fc, and ActRIIB-ALK4-Fc
ActRIIA-Fc was purchased from R&D (Cat. No. 340-RC2-100, Minneapolis, MN). ActRIIB-Fc was expressed and purified from Chinese hamster ovary (CHO) cells as previously described (*Sako et al., 2010*; *Cadena et al., 2010*). Briefly, ActRIIB-Fc was isolated using affinity chromatography with Mab Select Sure Protein A (GE Healthcare, Waukesha, WI), followed by dialysis into 10 mM Tris, 137 mM NaCl, and 2.7 mM KCl, pH 7.2. ActRIIB-ALK7-Fc and ActRIIB-ALK4-Fc were designed and expressed as previously described in CHO DUKX cells through the coexpression of two plasmids, each containing a receptor ECD (ActRIIB or ALK4) fused to a modified human IgG1 Fc domain (*Li et al., 2021*; *Kumar et al., 2021*). Purification was performed through protein A MabSelect SuRe chromatography (Cytiva), then eluted with glycine at low pH. The resulting sample was further purified over a Ni Sepharose 6 fast flow column (Cytiva) followed by an imidazole elution gradient, an ActRIIB affinity column and ultimately, a Q Sepharose column (Cytiva). ALK7-Fc production was performed as described previously (*Kumar et al., 2021*).

### Antibodies neutralizing ActRIIA, ActRIIB, ActRIIA/ActRIIB, and ALK7
An anti-ActRIIA antibody was obtained through phage-display technology, while anti-ActRIIB, anti-ActRIIA/ActRIIB and ALK7 antibodies were generated commercially by Adimab using their antibody discovery platform. The first three antibodies were expressed from stable CHO pools, while anti-ALK7 was transiently expressed in ExpiCHO cells (Thermo Fisher). The conditioned media (CM) was purified over Mab SelectSure Protein A (Cytiva) followed by ion-exchange chromatography.

### ActA, ActB, and Inhibin A
Mature, recombinant ActA and ActB were prepared as previously described (*Goebel et al., 2019a*; *Walker et al., 2017*). Briefly, ActA (pAID4T) was expressed in CHO DUKX cells. CM of ActA was then mixed with a proprietary affinity resin made with an ActRIIA-related construct (Acceleron). The resin was then lowered to pH 3 to dissociate the propeptide-ligand complex. Following this, the pH was raised to 7.5 and the resin was incubated for 2 hr at room temperature. ActA was eluted with 0.1 M glycine pH 3.0, which was concentrated over a phenyl hydrophobic interaction column (Cytiva) and eluted with 50% acetonitrile/water with 0.1% trifluoroacetic acid (TFA). Lastly, ActA was further

purified by HLPC over a reverse phase C4 column (Vydac) with a gradient of water/0.1% TFA and acetonitrile/0.1% TFA. Expression of ActB was performed through the use of a previously generated CHO-DG44 stable cell line (*Walker et al., 2017*). CM was initially clarified over an ion exchange SP XL column (Cytiva) in 6 M urea, 25 mM MES, 50 mM Tris pH 6.5. The flow-through was then adjusted to 0.8 M NaCl and applied to a Phenyl Sepharose column (Cytiva) and ActB was eluted by decreasing the NaCl through a gradient. Lastly, ActB was purified by reverse phase chromatography (C18, Cytiva) and eluted similarly to ActA. Recombinant human ActB used in the assays involving differentiated adipocytes was purchased from R&D (Cat. No. 659-AB-005). Inhibin A was produced and purified as previously described (*Pangas and Woodruff, 2002*).

## ActAC and ActC

Mature recombinant human ActAC and ActC were purchased from R&D (Cat. No. 4879-AC and 1629-AC, respectively). Antibodies used include: ActA (AF338, R&D); ActC (MAB1639, R&D); Goat (PI-9500, Vector Laboratories) and Mouse (DC02l, Calbiochem). ActC WT (IQP) and ActC Gln267Ala (IAP) were expressed transiently in HEK293T cells using a construct with an optimized furin cleavage site. The CM was then adjusted to 0.8 M NaCl and applied to a Phenyl Sepharose column (Cytiva) followed by elution with low NaCl. Lastly, ActC was then purified using reverse phase chromatography (C18, Cytiva) and eluted similarly to ActA and ActB.

## Fst-288 and Fstl3

Both Fst-288 and Fstl3 were produced as previously described (*Stamler et al., 2008*). Fst-288 was expressed from a stably transfected CHO cell line and purified from CM by binding to a heparin-Sepharose column (abcam) in 100 mM NaBic pH 8 and 1.5 M NaCl, with a low salt gradient to elute followed by cation exchange over a Sepharose fast flow (Cytiva) in 25 mM HEPES pH 6.5, 150 mM NaCl with a high salt elution gradient. Finally, Fst-288 was then purified over an HPLC SCX column in 2.4 mM Tris, 1.5 mM imidazole, 11.6 mM piperazine pH 6 with a high salt, high pH (10.5) gradient elution. Fstl3 was cloned into the pcDNA3.1/myc-His expression vector and expressed transiently in HEK293F cells. CM was harvested after 6 days and applied to His-affinity resin (Cytiva), followed by washing with a buffer of 500 mM NaCl, 20 mM Tris pH 8 and elution with 500 mM imidazole. Fstl3 was then subjected to SEC using a HiLoad Superdex S75 16/60 column (Cytiva) in 20 mM HEPES pH 7.5 and 500 mM NaCl.

## Luciferase reporter assays

Assays using the HEK-293-(CAGA)$_{12}$ or BRITER luciferase reporter cells were performed in a similar manner as described previously (*Goebel et al., 2019a*; *Walker et al., 2017*; *Goebel et al., 2022*; *Nolan et al., 2013*). Specifically, cells were plated in a 96-well format ($3 \times 10^4$ cells/well) and grown for 24 hr. For standard EC50 experiments (*Figure 1B and C*), growth media was removed and replaced with serum free media supplemented with 0.1% BSA (SF$^{BSA}$ media, Thermo Fisher) and the desired ligand, where a twofold serial dilution was performed with a starting concentration of 160 nM (ActA and ActAC, [CAGA)$_{12}$], or 4.96 nM (ActC, [CAGA]$_{12}$ and ActA, ActAC, ActC, and BRITER). Incubation was performed for 18 hr, cells were then lysed and assayed for luminescence using a Synergy H1 hybrid plate reader (BioTek - Winooski, VT). For the assays featuring transfections of ALK4$_{st}$, ALK5$_{st}$, or ALK7$_{st}$, a total of 50 ng DNA (10 ng type I receptor, 40 ng empty vector) was transfected using Mirus LT-1 transfection reagent at 24 hr postplating. Each receptor construct contains a single point mutation (pRK5 rat ALK5 S278T (ST), pcDNA3 rat ALK4 S282T, pcDNA4B human ALK7 S270T) conferring resistance to the inhibitory effects of the small molecule SB-431542. Media was then removed and replaced with SF$^{BSA}$ media with 10 μM SB-431542 and the desired ligand for 18 hr. For the experiments featuring ActRIIA-Fc, ActRIIB-Fc, the neutralizing antibodies, Fst288 or Fstl3, these proteins were added to the ligands and incubated for 10 min prior to addition to cells. The cell lines used for luciferase assays were purchased from ATCC and tested negative for mycoplasma contamination. The activity data were imported into GraphPad Prism and fit using a non-linear regression to calculate the EC$_{50}$ or IC$_{50}$.

## Surface plasmon resonance (SPR) studies

SPR experiments were carried out in HBS-EP + buffer (10 mM HEPES pH 7.4, 500 mM NaCl, 3.4 mM EDTA, 0.05% P-20 surfactant, 0.5 mg/ml BSA) at 25 C on a Biacore T200 optical biosensor system

(Cytiva). Fc-fusion constructs of each receptor were captured using either a series S Protein A sensor chip (GE Healthcare) or a series S CM5 sensor chip (GE Healthcare) with goat antihuman Fc-specific IgG (Sigma-Aldrich, Saint Louis, MO) immobilized with a target capture level of ~70 RU. Experiments with ActRIIA-Fc (ActA, ActAC, ActB), ActRIIB-Fc (ActA, ActAC, ActB), and ActRIIB-ALK4-Fc (ActA, ActAC, ActC) were performed with the former chip while experiments coupling ActRIIA-Fc (ActC), ActRIIB-Fc (ActC), and ActRIIB-ALK7-Fc (ActA, ActAC, ActC, ActB) were performed with the latter chip methodology. An 8-step, twofold serial dilution was performed in the aforementioned buffer for each ligand, with an initial concentration of 10 nM (for Activin C, a 10-step, twofold serial dilution beginning at 150 nM was performed, for Activin AC, a 9-step, twofold serial dilution starting at 20 nM was performed). Each cycle had a ligand association and dissociation time of 300–600 s, respectively. The flow rate for kinetics was maintained at 50 uL/min. SPR chips were regenerated with 10 mM Glycine pH 1.7. Kinetic analysis was conducted using the Biacore T200 evaluation software using a 1:1 fit model with mass transport limitations (red lines). Each binding experiment was performed in duplicate, fit individually and then averaged.

## Structural modeling and alignments

The model of ActC was built with Swiss-model using several ActA structures as templates: PDB codes 1S4Y (ActA:ActRIIB), 2ARV (unbound ActA), 2B0U (ActA:Fs288), 5HLZ (Pro-ActA) and 7OLY (ActA:ActRIIB:ALK4) (*Goebel et al., 2022*; *Thompson et al., 2005*; *Waterhouse et al., 2018*; *Greenwald et al., 2004*; *Wang et al., 2016*; *Harrington et al., 2006*). A consensus was observed in the overall structure, particularly at the type II interface and IAP motif. Ultimately, the model built from 7OLY was used for the comparison in *Figure 5*, as it is the most complete ActA-receptor complex, and all images and alignments were performed in PyMol (The PyMol Molecular Graphics System, Schrödinger, LLC, NY).

## Adipocyte isolation, differentiation, and treatment for western blot

Adipocyte stem cells were isolated, cultured, and differentiated as previously described (*Bowles et al., 2014*). Briefly, inguinal adipose tissue was harvested aseptically from male mice (3–4 weeks old) and placed in sterile PBS, followed by mincing and collagenase digestion (1 mg/ml) for 1 hr at 37 °C. Then, the digestion was filtered through a 70 μm mesh and centrifuged to separate the SVF. Following aspiration, the SVF was resuspended in DMEM supplemented with 10% FBS and Pen-strep-amphotericin (Wisent Inc cat. No: 450–115-EL, Saint-Jean-Baptiste, Canada) and plated in a 6-well format at ~320,000 cells/well. Following expansion over 4 days, cells were differentiated over the course of 4 days using a solution of 5 μM dexamethasone, 0.5 mM 3-isobutyl-1-methylxanthine, 10 μg/ml insulin and 5 μM Rosiglitazone. Adipocytes were then maintained for six additional days prior to experimentation in DMEM/FBS + insulin. 3T3-L1 cells were differentiated to adipocytes following ATCC recommended protocol. Briefly, 3T3-L1 cells were differentiated over the course of 4 days using a solution of 1 μM dexamethasone, 0.5 mM 3-isobutyl-1-methylxanthine and 1 μg/ml insulin. 3T3-L1 chemically-induced adipocytes were then maintained for six additional days prior to experimentation in DMEM/FBS + insulin. Differentiated adipocytes from SVF or 3T3-L1 cells were then starved in serum-free media for 1 hr, after which they were treated with serum-free media containing ActA, ActB, or ActC (2 nM) for 1hr ± Fst-288 (800 ng/ml). In another set of experiments, differentiated-SVF cells were treated with ActA, ActB, or ActC (2 nM) for 1hr ± anti-ALK7 antibody (30 μg/ml). Concentrations were selected based on in vitro cell-based assays. At the end of the treatments, cells were lysed using RIPA buffer containing protease inhibitors and western analysis was performed using anti-pSMAD2 (Cell Signaling, 138D4, Danvers, MA) or anti-SMAD2/3 antibodies (Millipore, 07–408, Burlington, MA). 3T3-L1 cells were purchased from ATCC and tested negative for mycoplasma contamination.

## Adipocyte RNA extraction

Cells were collected in TRIzol and RNA was extracted following the manufacturer's protocol (Zymo Research). Total RNA from SVF or 3T3-L1 adipocyte-differentiated cells (at day 10 of differentiation) (200 ng) was reverse transcribed using (MMLV) reverse transcriptase following the manufacturer's protocol (Promega, Madison, WI). Expression of genes encoding the *Pparg2*, *Cebpa*, and *Pnpla2* was analyzed in duplicate qPCR reactions using EvaGreen Master mix (ABMMmix-S-XL; Diamed) on a Corbett Rotorgene 6000 instrument (Corbett Life Science, Mortlake, NSW, Australia). Gene

expression was determined relative to the housekeeping gene *Rpl19* using the 2-ΔΔCt method (*Livak and Schmittgen, 2001*). Primer sequences are listed in *Supplementary file 3*.

## Adipocyte images and Oil Red O staining

Before and after day 10 of differentiation, adipocyte images were acquired with an Axiocam 506 mono camera (Zeiss, White Plains, NY) using ZEN 2.3 pro (Zeiss) software. For Oil Red O (ORO) staining, cells were washed in PBS and fixed in 10% formalin buffered solution for 10 min. After fixation, cells were washed in 60% isopropanol and stained in an ORO solution (2:3 v/v $H_2O$: isopropanol, containing 0.5% ORO, Sigma O0625) for 1 hr. After staining, cells were washed in PBS and dye from lipid droplets was extracted by adding pure isopropanol for 10 min in a rotor shaker. Dye per well was quantified by absorbance at 500 nm in EZ Read 2000 microplate reader (Biochrom, Holliston, MA). After washing with PBS, cells were digested using a 0.25% Trypsin solution in PBS-EDTA for 24 hr at 37°C. DNA was quantified using a NanoDrop, and cell lipid content was normalized by the corresponding cell DNA content per well.

## Additional information

### Competing interests

Elitza Belcheva, Roselyne Castonguay, Ravindra Kumar: Past employee of Acceleron Pharma and is now an employee of Merck Sharp & Dohme Corp., a subsidiary of Merck & Co., Inc. The other authors declare that no competing interests exist.

### Funding

| Funder | Grant reference number | Author |
|---|---|---|
| National Institute of General Medical Sciences | R35 GM134923 | Thomas B Thompson |

The funders had no role in study design, data collection and interpretation, or the decision to submit the work for publication.

### Author contributions

Erich J Goebel, Conceptualization, Data curation, Formal analysis, Investigation, Methodology, Project administration, Validation, Visualization, Writing - original draft, Writing – review and editing; Luisina Ongaro, Conceptualization, Data curation, Formal analysis, Investigation, Methodology, Visualization, Writing - original draft, Writing – review and editing; Emily C Kappes, Investigation, Methodology; Kylie Vestal, Data curation, Investigation; Elitza Belcheva, Investigation, Visualization; Roselyne Castonguay, Conceptualization, Data curation, Formal analysis, Methodology, Visualization, Writing – review and editing; Ravindra Kumar, Conceptualization, Formal analysis, Project administration, Resources; Daniel J Bernard, Conceptualization, Formal analysis, Methodology, Visualization, Writing – review and editing; Thomas B Thompson, Conceptualization, Formal analysis, Funding acquisition, Investigation, Methodology, Project administration, Resources, Supervision, Visualization, Writing – review and editing

### Author ORCIDs

Erich J Goebel ⓘ http://orcid.org/0000-0001-5549-9425
Emily C Kappes ⓘ http://orcid.org/0000-0002-3063-3373
Daniel J Bernard ⓘ http://orcid.org/0000-0001-5365-5586
Thomas B Thompson ⓘ http://orcid.org/0000-0002-7041-5047

### Ethics

Male wild-type mice were euthanized at 3-4 weeks old in accordance with institutional and federal guidelines and approved by the McGill University and Goodman Cancer Centre Facility Animal Care Committee (protocol 5204).

### Decision letter and Author response

Decision letter https://doi.org/10.7554/eLife.78197.sa1

Author response https://doi.org/10.7554/eLife.78197.sa2

## Additional files

### Supplementary files
• Supplementary file 1. Linear regression analysis. The activity data corresponding to *Figures 1B, D, 4C and 5E* were imported into GraphPad prism and fit using a non-linear regression to calculate the $EC_{50}$ or $IC_{50}$ with standard error was calculated from triplicate experiments.

• Supplementary file 2. SPR Kinetic Analysis.

• Supplementary file 3. qPCR primer sequences.

• Transparent reporting form

• Source data 1. Luciferase, SPR and raw image data for all figures.

### Data availability
All data generated or analysed during this study are included in the manuscript and supporting file.

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
