## [Editor Report]

The function of Activin C is poorly understood. The authors show that Activin C stimulates SMAD2/3 signaling via the type I receptor ALK7. However, Activin C binds the cognate Activin type II receptor with a lower affinity than Activin A or Activin B and is resistant to the extracellular antagonist Follistatin. Collectively, these data clarify the biological activity of Activin C and provide an important foundation for further research on Activin signaling.

---

## [Decision Letter]

**Decision letter after peer review:**

Thank you for submitting your article "The orphan ligand, Activin C, signals through activin receptor-like kinase 7" for consideration by *eLife*. Your article has been reviewed by 2 peer reviewers, one of whom is a member of our Board of Reviewing Editors, and the evaluation has been overseen by Marianne Bronner as the Senior Editor. The following individual involved in review of your submission has agreed to reveal their identity: Andrew P Hinck (Reviewer #2).

Essential revisions:

Overall, there are no major technical concerns regarding the results the authors report, though a few points are listed below:

1. Since the authors have fit their cell-based reporter data to dose response curves, they should report the fitted EC50 values (and errors), and other parameters such as slope, in a supplementary data table.

2. Unlike efforts that were made to understand and experimentally evaluate the type II receptor binding preferences of ActC (compared to ActA and ActB), no experimental effort was made to understand and evaluate the underlying basis of ActC's unique type I receptor binding preference; in the absence of this (which is understandable, given the much greater complexity compared to the situation with type II receptor binding), it would nonetheless be helpful for the readers if the authors could add a supplementary figure showing the main interactions between activin class ligands and their type I receptors and as best as possible compare the corresponding positions to those in Alk7 and ActC.

3. It is stated in the Discussion, that ActC may bind to the type I and type II receptors cooperatively; however, it is unclear what specifically is meant by this; are they suggesting that there is positive allostery with one receptor enhancing the binding affinity of the other (which would be unique to this type I-type II receptor pair in the TGFbeta family) or are they suggesting avidity effects are at play and an enhanced affinity is observed due to the simultaneous binding of both type I and type II receptors at non-overlapping (and non-interacting sites) (which would not be unique to this type I-type II receptor pair in the TGFbeta family)?

As evidence for "cooperatively, they cite the higher affinity of the Alk7-ActRIIB heterodimer for ActC compared to either receptor alone; in my mind, the simplest explanation for this is effects arising from avidity, not positive allostery; to conclude positive allostery would require performing an experiment in which the binding of one of the receptors is measured in the absence and presence of a saturating concentration of the other receptor. In light of the potential confusion surrounding this point, the authors should clarify what they mean by cooperativity.

4. In order to determine to what extent ActC IAP recapitulates ActRII binding affinity (compared to ActA), the authors should fit the data shown in Figure 5F and report the values in Supplementary Table 1; they should also mention this value in the Results section of the text.

5. There are two typos in the Introduction – on line 41, 'heterodimers have can form" is non-sensical on line 53 a reference is missing.

6. It the Methods section, it is unclear what the "Adimab platform" is; this should be better described/referenced.

7. In light of the observed antagonism of ActC by InhA and reported ActC receptor binding properties, is there any potential role of ActC in regulating endocrine function?

---

## [Author Response]

Essential revisions:Overall, there are no major technical concerns regarding the results the authors report, though a few points are listed below:1. Since the authors have fit their cell-based reporter data to dose response curves, they should report the fitted EC50 values (and errors), and other parameters such as slope, in a supplementary data table.

We agree that reporting these values would add to the presentation/analysis of the data – as such we have included Supplementary file 1 (and renumbered the previous tables) to include EC50 values from Figures 1 and 5 and IC50 values from Figure 4 along with the standard error. We have also modified the Results section to better capture the analysis performed, which now reads:

“The activity data were imported into GraphPad Prism and fit using a non-linear regression to calculate the EC_50_ or IC_50_.”

2. Unlike efforts that were made to understand and experimentally evaluate the type II receptor binding preferences of ActC (compared to ActA and ActB), no experimental effort was made to understand and evaluate the underlying basis of ActC's unique type I receptor binding preference; in the absence of this (which is understandable, given the much greater complexity compared to the situation with type II receptor binding), it would nonetheless be helpful for the readers if the authors could add a supplementary figure showing the main interactions between activin class ligands and their type I receptors and as best as possible compare the corresponding positions to those in Alk7 and ActC.

We agree that a visual representation of the interfaces described in the discussion would aid the reader in understanding the nuances of Activin-Type I receptor specificity. As such, we have generated a supplemental figure (Figure 7—figure supplement 2) to accompany our description of the potential contacts of the ActC/ALK7 interface.

3. It is stated in the Discussion, that ActC may bind to the type I and type II receptors cooperatively; however, it is unclear what specifically is meant by this; are they suggesting that there is positive allostery with one receptor enhancing the binding affinity of the other (which would be unique to this type I-type II receptor pair in the TGFbeta family) or are they suggesting avidity effects are at play and an enhanced affinity is observed due to the simultaneous binding of both type I and type II receptors at non-overlapping (and non-interacting sites) (which would not be unique to this type I-type II receptor pair in the TGFbeta family)?As evidence for "cooperatively, they cite the higher affinity of the Alk7-ActRIIB heterodimer for ActC compared to either receptor alone; in my mind, the simplest explanation for this is effects arising from avidity, not positive allostery; to conclude positive allostery would require performing an experiment in which the binding of one of the receptors is measured in the absence and presence of a saturating concentration of the other receptor. In light of the potential confusion surrounding this point, the authors should clarify what they mean by cooperativity.

We agree that this statement could be confusing/misleading, especially using the term “cooperative” without direct evidence. To clarify, we believe that the ActC’s receptor binding might be driven by avidity given our data with the ALK7-ActRIIB heterodimer. We have reworded the statement in the conclusion to better reflect this, which now reads:

“One possible explanation is avidity from binding contribution both the type I and type II receptors are needed for ActC to signal.”

4. In order to determine to what extent ActC IAP recapitulates ActRII binding affinity (compared to ActA), the authors should fit the data shown in Figure 5F and report the values in Supplementary Table 1; they should also mention this value in the Results section of the text.

We agree that a fit would enhance the reader’s ability to determine the similarity between ActC IAP and ActA binding to ActRII and have added a steady state analysis in Figure 2—figure supplement 1 with a steady state analysis of ActC-IAP binding to ActRIIA-Fc (Figure 5F) and reported the KD value (620pM) in Supplemental table 1 (now Supplementary file 2) and within the Results section of the text.

5. There are two typos in the Introduction – on line 41, 'heterodimers have can form" is non-sensical on line 53 a reference is missing.

We have corrected both of these typos accordingly.

6. It the Methods section, it is unclear what the "Adimab platform" is; this should be better described/referenced.

We have clarified our explanation of the platform, which now reads:

“…antibodies were generated commercially by Adimab using their antibody discovery platform.”

7. In light of the observed antagonism of ActC by InhA and reported ActC receptor binding properties, is there any potential role of ActC in regulating endocrine function?

While the logical next step of an ActC project is to test the potential biological effects of ActC more extensively, we don’t feel comfortable making statements regarding endocrine function currently.